# Effect of Fertilization on Yield and Quality of *Sisymbrium officinale* (L.) Scop. Grown as Leafy Vegetable Crop

**Marta Guarise [1],\*, Gigliola Borgonovo [2], Angela Bassoli [2] and Antonio Ferrante [1]**

[1]  Department of Agriculture and Environmental Science—Production, Landscape, Agroenergy, Università degli Studi di Milano, 20133 Milano, Italy

[2]  Department of Food, Environmental and Nutritional Science, Università degli Studi di Milano, 20133 Milano, Italy

\*  Correspondence: marta.guarise@unimi.it; Tel.: +39-0250316593

**Abstract:** *Sisymbrium officinale* is a wild Brassicaceae species that is known for its use in the alleviation of vocal inflammatory states. Since this species is particularly rich in bioactive compounds, there is an interest for developing cultivation protocols to use this plant as a leafy vegetable harvested at the 13th BBCH growth stage. Two wild populations of *S. officinale* (L.) Scop., denominate MI (Milan) and BG (Bergamo), have been used, and three different levels of nutrients (g/m$^2$) have been provided such as 13 N, 7 P$_2$O$_5$, 8 K$_2$O (100%); 9 N, 5 P$_2$O$_5$, 5.5 K$_2$O (70%); and 6.5 N, 3.5 P$_2$O$_5$, 4.0 K$_2$O (50%). The effects of different fertilization levels were evaluated on the yield, leaf pigments (chlorophylls, carotenoids), nitrate concentration, sugars content, and on the antioxidant compounds such as anthocyanins, total phenols and glucosinolates (GLS). Plant stress monitoring was performed by measuring the chlorophyll *a* fluorescence. Results indicated that yield was not affected and ranged from 0.18 to 0.47 kg/m$^2$, and differences were not statistically significant for chlorophylls, carotenoids, and total sugars content. Nitrate concentrations were higher in the BG wild population (4388.65 mg/kg FW) during the second cultivation cycle and lower (1947.21 mg/kg FW) in the same wild population during the first cultivation cycle, both at the 100% fertilization level. Total phenols and anthocyanins were higher in the highest fertilization rate in the MI wild population during the second cycle. The GLS were higher in the lowest fertilization rate in the BG wild population (19 µmol/g FW) grown at the fertilization level of the 50%, and significant differences were observed in the second cycle. In conclusion, the 50% fertilization level can be considered the most suitable for our experimental conditions.

**Keywords:** baby leaf; NPK fertilization; nitrate; *Sisymbrium officinale*; wild species; glucosinolates

## 1. Introduction

Wild foods constitute an important component of people's diet around the world. Wild edible plants (WEPs) are a subcategory of cultivated plants that include neglected crops. Many of them are classified as weeds, species that can easily and promptly grow and reproduce in disturbed land, such as urban and agricultural areas [1], and are available in various environments and agroecosystems.

WEPs have the potential to improve nutrition and diets, diversify farm production, and improve the resilience of crops [2].

Generally, wild plants contain many secondary metabolites or bioactive compounds, such as polyphenols, polysaccharides, and terpenoids, which act as "nutraceuticals." For these reasons, they can be used as food-medicine and also be named as functional foods [3]. On the other hand, WEPs may accumulate high levels of toxins from the environment where they grow [2].

A total of 67 WEPs, belonging to 20 different families, have been reported in literature for their health proprieties such as laxative, diuretic, digestive, antitussive (cough), hypotensive, tonic, sedative, and hypoglycaemic [4]. The diversity of wild harvested plants includes hypo and epigeal parts, such as roots, shoots and leafy greens; reproductive parts, such as berries and other fleshy fruits, grains, nuts and seeds; and mushrooms, lichens and algae [1]. In order to protect this diversity of flora, there are limitations and prohibition on the harvest of certain wild species [5].

Wild plants are commonly use in traditional medicine, and there is great interest regarding the understanding of the agro-industrial potential of their chemical and nutraceutical characteristics to increase their commercial distribution and necessary investments [6].

Nowadays, WEPs are underutilized: The act of collecting wild species requires more time and is strongly influenced by seasonality—more than buying food from the supermarket. Nevertheless, changes in nutrition and the exploitation of natural remedies for self-medication must be considered. A renewed attention in such natural resources has also been confirmed by the gastronomic élite searching for new culinary stimuli and experiences [7]. Furthermore, the evaluation of non-traditional plants may bring consumers towards new compounds with biological activity [8], unusual colors, and flavors [9].

*Sisymbrium officinale* (L.) Scop. (synonym *Erisymum officinale*), also known as hedge mustard in English, is mostly widespread in Europe, Western Asia and North Africa. This plant is described as a wild species present in all Italian Regions from 0 up to 1000 a.s.l. (above sea level), and it is common in bare ground, roadsides, and edges of uncultivated fields. *S. officinale* is an herbaceous plant that belongs to the Brassicaceae family described as annual or, in some cases, biennial with a reddish-violet erect trunk, upper leaves with a dentate shape and a linear racemose inflorescence. Small flowers present yellow petals, and the fruit is a tiny siliqua containing from 10 to 20 small seeds. Siliqua pods are pubescent, and once they reach maturity, usually between July and August, they release seeds [10–12].

*S. officinale* is commonly known as the "singers' plant" for its apparently effects on vocal tract and upper respiratory trait diseases. Hedge mustard's flowers and leaves are commonly used as a traditional remedy for the treatment of sore throats, coughs and hoarseness [13,14]. Its effect on alleviating vocal trait in a small group (104 patients) with different diseases of the vocal tract, in the absence of major diseases, has been recently reported [15].

Brassicaceae is one of the most important families for horticultural production in the Mediterranean area, and it is generally known to be rich in sulphur compounds [16]. Glucosinolates (GLS) are the principal compounds of the secondary metabolism found in Cruciferous crops. GLS are sulfur-rich, anionic natural products that upon hydrolysis by myrosinase, an endogenous thioglucosidase that produces, among others, isothiocyanates, thiocyanates, and nitriles. These hydrolytic products have many different biological activities for plants—as crop protection compounds and biofumigants in agriculture to suppress pathogens, nematodes, and weeds—and for humans—such as cancer-preventing agents and flavor compounds [17].

In agricultural systems, nutritional needs and fertilization are strongly influenced by the complex interaction between the genetic characteristics of horticultural species/cultivars and the cultivation environment, which includes soil fertility and agronomic practices. For that reason, the aim of proper fertilization management is to fix the dose, time, formulations, and application methods of mineral elements to reach high qualitative and quantitative productions in respect to the environment and human health [18]. Among all the mineral elements, nitrogen is the most required by plants and is absorbed in two forms: Ammonium ($NH_4^+$) and nitrate ($NO_3^-$). In plants, nitrate is accumulated in vacuoles or reduced into nitrite and then incorporated into amino acids. Moreover, it can be useful as a chemical osmoregulator, especially in sugar deficiency, during low light intensity periods [19].

The main sources of nitrate intakes are drinking water, cured meat, cheese, grain, and vegetables, all of which constitute the principal source of this nutritive element in everyday diet.

Genetic, environmental, and agricultural factors affect $NO_3$ uptake and accumulation in vegetable tissues.

Nitrate is relatively harmless, but it can become dangerous when it is reduced into nitrite ($NO_2^-$) and N-nitroso compounds. When nitrite reacts with haemoglobin (oxyHb), it will form methaemoglobin (metHb) and $NO_3^-$, causing methaemoglobinaemia, a potentially fatal condition that can lead to cyanosis, asphyxia, and suffocation. Furthermore, nitrite may react with amines or amides to form carcinogenic N-nitroso compounds [20].

Long term health risks of high nitrate intake are still under debate, and even if there is contradictory evidence regarding its potential damaging effects, the European Food Safety Authority has defined the acceptable daily intake (ADI) for a 60 kg adult as 222 mg $NO_3$ per day [21], while EU (European Union) Regulation 1258/2011 fixes the maximum nitrate levels permitted in foodstuff. The maximum nitrate level allowed in rocket salad, another *Brassica* species, is fixed at 6000–7000 $NO_3$/kg fresh matter (FM), depending on harvest time [22].

As human exposure to nitrate is mainly exogenous and these compounds are highly accumulated especially in leafy vegetables, such as *S. officinale*, the reduction of $NO_3$ levels in vegetable crops must be achieved with an appropriate management of N fertilization [23].

Furthermore, *Brassica* species have always been associated with salutary effects due to the presence of Vitamins C and E, carotenoids, phenols, and flavonoids, which are generally considered as bioactive compounds. Flavonoids have an antioxidant capacity that scavenges free-radicals and reactive oxygen species (ROS). Genetic factors, harvesting time, growing conditions (such as soil state, nutritional deficiency, or even temperature and radiation), and post-harvest storage may induce variation in the antioxidant contents [24,25]. The antioxidant ability, such as radical scavenging, of *S. officinale* dry extract and its polyphenolic components have been studied to confirm the traditional use of this plant by smokers to restore vocal cord function [26].

In today's modern society, the minimally processed ready-to-eat industry is rapidly evolving to add new variety of crops with a particular appeal, taste, and texture to offer consumer products rich in beneficial compounds. The Asteraceae and Brassicaceae families are mostly used in the ready-to-eat industrial production [27].

Therefore, the purpose of the present study was to evaluate the effect of three different fertilization levels on the growth of two wild population of *S. officinale* in order to evaluate the possibility to cultivate this species, commonly used only for medical purposes, as a potential new leafy vegetable.

## 2. Materials and Methods

### 2.1. Plant Materials

Two wild populations of *Sisymbrium officinale* (L.) Scop. named MI (Milan) and BG (Bergamo), obtained from a controlled seed reproduction at Fondazione Minoprio (Como, Italy) during summer in 2017, were grown under controlled conditions in a greenhouse at the Faculty of Agricultural and Food Science of the University of Milan (45°28'33.2" N, 9°13'38.6" E) (Figure S1)

The seeds of the two wild populations were sown twice, 23 January 2018 for the first cultivation cycle and 2 March 2018 for the second cycle. Sowing was performed in polystyrene panels using common horticultural fertilized substrate, keeping the two wild populations separated.

Plantlets were transplanted and grown using complete substrate (Vigorplant Italia Srl, Fombio (LO), Italy) for horticultural cultivation (containing the following components: 21% Baltic peat, 22% dark peat, 26% Irish peat, 13% volcanic peat, 18% calibrated peat) with pH 6.5 in 14 cm diameter plastic pots of 2 L volume. The transplant was performed the 15th March 2018 for the first cycle and the 16th May for the second cultivation cycle using 18 plantlets each from the MI and BG wild populations (Table S1).

Plants were supplied with three different levels of NPK granular fertilizer (14:7:17) in both experimental periods and were watered every day, keeping the optimal water conditions.

In the definition of the different levels of fertilization, rapeseed (*Brassica napus* L.) was used as plant model. Six plants of each wild population were fertilized with 4 g/pot (level 100%), six plants

were fertilized with 2.8 g/pot (level 70%), and the last six plants were fertilized with 2 g/pot (level 50%) (Table S2).

The amount of fertilizers corresponds to:

$$-100\% = 13 \text{ g/m}^2 \text{ N}; 7 \text{ g/m}^2 \text{ P}_2\text{O5}; 8 \text{ g/m}^2 \text{ K}_2\text{O} \tag{1}$$

$$-70\% = 9 \text{ g/m}^2 \text{ N}; 5 \text{ g/m}^2 \text{ P}_2\text{O}_5; 5.5 \text{ g/m}^2 \text{ K}_2\text{O} \tag{2}$$

$$-70\% = 9 \text{ g/m}^2 \text{ N}; 5 \text{ g/m}^2 \text{ P}_2\text{O}_5; 5.5 \text{ g/m}^2 \text{ K}_2\text{O} \tag{3}$$

$$-50\% = 6.5 \text{ g/m}^2 \text{ N}; 3.5 \text{ g/m}^2 \text{ P}_2\text{O}_5, 4.0 \text{ g/m}^2 \text{ K}_2\text{O} \tag{4}$$

Harvest was performed at the 13 BBCH (Biologische Bundesanstalt, Bundessortenamt and Chemical insustry) growth stage for each cultivation cycle, on 14 May 2018 for the first cultivation cycle and on 26 June 2018 for the second cycle. Sampling was randomized—plants were casually chosen from each pot.

The fresh and dry biomass of the two wild populations of *S. officinale* were measured to evaluate the total production of leaves. Finally, carotenoids, phenols, anthocyanins, nitrate and total sugars, commonly used to define quality parameters in leafy vegetables, were measured.

### 2.2. Non-Destructive Determination: Chlorophyll a Fluorescence Determination

A non-destructive analysis was carried out on leaf tissue for the agronomic characterization of two *S. officinale* wild populations at the end of cultivation cycle. Chlorophyll *a* fluorescence was measured using a portable fluorometer (Handy PEA, Hansatech Instruments Ltd, Kings Lynn, UK). Leaves were dark-adapted using leaf clips; after 30 min, a rapid pulse of high-intensity light of 3000 $\mu$mol m$^{-2}$ s$^{-1}$ (600 W m$^{-2}$) was applied to the leaf, thus inducing fluorescence.

Fluorescence parameters were calculated automatically by the used device, such as Fv/Fm, the variable fluorescence to maximum fluorescence. Starting from these parameters, JIP analyses was performed to determine the following indexes: Performance index (PI); the dissipation of energy per cross-section (DIo/RC), and the density of reaction center at the P stage (RC/CSm).

### 2.3. Destructive Determination

For the evaluation of qualitative characteristic of the two wild hedge mustard populations (MI and BG) related to the three different fertilization levels, about 1 g of fresh leaves for each sample was collected at the end of each cultivation cycle. To prevent leaf tissue's degradation, each sample was conserved at −20 °C after the collection.

Plant fresh matter (FM) and dry matter (DM) were measured at the end of each cultivation cycle—in the middle of May for the first cycle and at the end of June for the second cycle—in order to evaluate the yield.

#### 2.3.1. Total Chlorophylls and Carotenoids Measurements

Total chlorophylls and carotenoids were extracted from 20–30 mg fresh leaves (disks of 5 mm diameter) using 5 mL of methanol (99.9%) as a solvent and kept in dark cold room at 4 °C for 24 h. Quantitative chlorophyll determinations were carried out immediately after extraction. Absorbance readings were measured at 665.2 and 652.4 nm for chlorophyll pigments and 470 nm for total carotenoids. Total chlorophylls and carotenoid concentrations were calculated by Lichtenthaler's formula [28].

#### 2.3.2. Phenolic Index and Anthocyanins

The Phenolic Index in leaf tissue was determined spectrophotometrically by the direct measurement of the leaf extract absorbance at 320 nm. About 20–30 mg of fresh leaf tissue (disk of 5 mm diameter)

was weighed, and 3 mL methanolic HCl (1%) were added. After overnight incubation, the supernatant was read at 320 nm. The values were expressed as mg/100 g FM.

Anthocyanin content was determined spectrophotometrically. Samples of 20–30 mg of fresh leaf (disks of 5 mm diameter) were extracted using 3 mL of methanolic HCl (1%). Samples were incubated overnight at 4 °C in darkness. The concentration of cyanidin-3-glucoside equivalents was determined spectrophotometrically at 535 nm using an extinction coefficient ($\varepsilon$) of 29,600 [29].

### 2.3.3. Nitrate Determination

Nitrate concentration was measured by the salicyl sulfuric acid method [30]. Samples for nitrate were collected once at the end of each cultivation cycle when plants reached the 13 BBCH growth stage.

About 1 g of fresh leaves was ground in 5 mL of distilled water. The extracts were centrifuged at 4000 rpm for 15 min. After centrifugation, the supernatant was collected for colorimetric determinations. Twenty µL of samples were collected, and 80 µL (5% w/v) of salicylic acid in concentrated sulfuric acid and 3 mL of NaOH 1.5 N were added to them. Each sample was cooled, and absorbance was measured at 410 nm. Nitrate concentration was calculated by referring to the $KNO_3$ standard calibration curve.

### 2.3.4. Total Sugar Determination

Samples for total sugar analyses were collected twice during the experiment, as explained for the nitrate samples. To determinate total sugar levels in *S. officinale* fresh leaves, extracts were prepared as explained for the determination of nitrate levels.

Total sugars were determined according the anthrone's assay with a slight modification [31]. The reagent (anthrone) was prepared using 0.1 g of anthrone dissolved in 50 mL of 95% $H_2SO_4$, and it was left 40 min before use. After this period, 200 µL of extract were added to 1 mL of anthrone, put in ice for 5 min, and vortexed. Samples was heated at 95 °C for making the reaction. After 5 min of incubation, samples were cooled, and absorbance was performed at 620 nm. Total sugar concentration was calculated by referring to the glucose standard calibration curve.

### 2.3.5. Glucosinolates Determination

Samples for the evaluation of glucosinolate content were collected at the end of each cultivation cycle. GLS content was obtained from the fresh leaves of two wild populations of *S. officinale* following the method described by the French pharmacopeia, with slightly differences [32]. GLS content was estimated spectrophotometrically according to Mawlong [33]. Three mL of 2 mM sodium tetrachloropalladate (obtained with 58.8 mg Sodium tetrachloropalladate, 170 µL concentrated HCl, and 100 mL double distilled water) and 0.3 mL of distilled water were added to 100 µL of extract. One hour of incubation was necessary to allow the formation of a complex among GLS and the palladate reagent, which led to a shift from light brown to dark, depending upon the GLS concentration.

After this period, absorbance was measured at 425 nm, and a calibration curve was determined by referring to six standards of sinigrin with concentrations ranging from 0.50 to 2.00 mM. The total GSL content was expressed as µmol/g of fresh material (FM).

### 2.4. Statistical Analyses

The experimental design was organized as follow: Two wild populations (MI and BG); three different fertilization levels (100%, 70%, and 50%); six plants for each fertilization levels for a total number of 18 plants for the MI wild population and 18 for the BG wild population; and two cultivation cycles.

Four replicates ($n = 4$) were taken for each fertilization level of each wild population for the evaluation of chlorophyll a fluorescence, and three replicates ($n = 3$) were taken for each fertilization level of each wild population for the other qualitative parameters that we decided to consider.

Data from the two cultivation cycles were subjected to multifactor ANOVA, and differences were determined by an LSD post-test. The analysis was performed using Statgraphics software (Statgraphics, Centurion XV, v. 15.2.05).

Two-way ANOVA (fertilization doses and wild population) was also performed, and differences among means were determined using Tukey's post-test ($p < 0.05$). The number of replicate samples used in each analysis or measurement is reported in the legend of the figures or tables. Two-way ANOVA was performed using GraphPad Prism version 6.00 for Windows (GraphPad Software, La Jolla, CA, USA).

## 3. Results

### 3.1. Chlorophyll a Fluorescence Measurement

From the chlorophyll *a* fluorescence data (Table 1), four parameters were considered: Fv/Fm (maximum quantum yield of PSII), PI (Performance Index), DI0/RC (rate of energy dissipated by PSII per reaction center), and RC/CSm (active RCs per excited cross-section).

**Table 1.** Chlorophyll *a* fluorescence parameters in leaves of two *S. officinale* wild populations, MI (Milan) and BG (Bergamo). Data are means with standard errors ($n = 4$).

| Cycle | Fertilization Level | Wild Population | Fv/Fm | PI | DIo/RC | RC/CSm |
|-------|--------------------|-----------------|-------|-----|--------|--------|
| **I** | 100% | MI | 0.84 ± 0.002 | 2.51 ± 0.045 | 0.43 ± 0.003 | 2162.44 ± 434.5 |
| | | BG | 0.84 ± 0.006 | 2.33 ± 0.26 | 0.43 ± 0.02 | 2109.75 ± 113.73 |
| | 70% | MI | 0.84 ± 0.003 | 2.44 ± 0.2 | 0.44 ± 0.025 | 2068.96 ± 129.3 |
| | | BG | 0.84 ± 0.008 | 2.49 ± 0.37 | 0.46 ± 0.05 | 2210.5 ± 142.45 |
| | 50% | MI | 0.84 ± 0.003 | 2.33 ± 0.2 | 0.45 ± 0.02 | 1968.51 ± 110.03 |
| | | BG | 0.84 ± 0.004 | 2.28 ± 0.3 | 0.45 ± 0.03 | 2079.49 ± 195.4 |
| **II** | 100% | MI | 0.85 ± 0.004 | 4.56 ± 0.82 a | 0.31 ± 0.03 b | 3159.74 ± 369.6 a |
| | | BG | 0.83 ± 0.011 | 2.38 ± 0.5 ab | 0.46 ± 0.05 ab | 2085.37 ± 191.55 ab |
| | 70% | MI | 0.85 ± 0.002 | 4.66 ± 0.47 a | 0.32 ± 0.02 b | 3062.5 ± 222.07 ab |
| | | BG | 0.84 ± 0.003 | 2.88 ± 0.35 ab | 0.39 ± 0.03 ab | 2454.64 ± 213.75 ab |
| | 50% | MI | 0.84 ± 0.006 | 2.67 ± 0.46 ab | 0.42 ± 0.03 ab | 2388.09 ± 194.22 ab |
| | | BG | 0.83 ± 0.005 | 1.96 ± 0.38 b | 0.49 ± 0.04 a | 2008.36 ± 238.22 b |

Data were subjected to multifactor ANOVA and differences among wild populations and dates within a cycle were determined using LSD's test ($p = 0.05$).

The multifactor ANOVA of the Fv/Fm data revealed that only the means between the two wild populations were statistically different. The interactions among the three factors (fertilization level, wild population, and cycle) were not significantly different (Table S3).

No significant differences were found in the Fv/Fm ratio either in the first or in the second cultivation cycle, and values ranged between 0.83 and 0.85, which means that both the two wild populations of *S. officinale* did not show stress conditions at the end of each cultivation cycle.

Multifactor ANOVA analysis for the PI parameter showed that all factors had $p < 0.05$. The interaction between "wild population × cycle" was statistically different $p < 0.05$. The PI measured from plants grown with a 50% fertilization rate was statistically different from 75% and 100%. Additionally, wild populations and growing cycles were also significantly different (Table S4).

The DIo/RC was statistically different for all parameters measured (Table S5). The interaction between "Fertilization × wild population" was not significant, while "wild population × cycle" and "fertilization × cycle" were statistically significant $p < 0.05$ (Table S5). The fertilization treatments were different between 50% and 100%, while the intermediate fertilization rate was not different among them. The multifactor ANOVA for the RC/CSm parameter showed significant statistic differences for all factors. Significant interaction was only found for the "wild population × cycle." The RC/CSm data were significantly different between 50% and 100%, while the 75% fertilization rate was not different

compared to the others. Significant differences were found between wild populations and growing cycles (Table S6).

During the second cultivation cycle, PI values were higher in the MI wild population at the fertilization levels of 100% and 70%, and they were statistically different than the BG wild population at the fertilization level of 50%. RC/CSm showed the same trend as the PI parameter, while for DIo/RC, higher values were recorded in the BG wild population at the lower fertilization rate, and the MI wild population showed lower values at the fertilization levels of 100% and 70%.

### 3.2. Yield, Total Chlorophylls, Carotenoids, Phenolic Index and Total Anthocyanins

The yield of the two wild populations, MI and BG, was measured for both cultivation cycles by harvesting the areal part of the plants at the 13 BBCH growth stage.

Multifactor ANOVA showed that yield was not significant for the fertilization and wild population factors, while the growing cycle factor was statistically significant for $p < 0.05$. Regarding DM data, the multifactor analysis showed that the interactions among factors were not significant. The significant differences were found for the wild population and growing cycle factors (Table S7).

Fresh matter (FM) ranged from 67.1 g/plant of 100% BG to 30 g/plant of the 50% MI wild population in the first cycle, and from 41.3 g/plant of 70% MI to 25.9 g/plant of the 50% BG wild population in the second cultivation cycle. The percentage of dry matter showed higher values in 50% BG and lower values in the 70% MI wild population for both cultivation cycles (Table 2).

**Table 2.** Fresh weight and % of dry matter in MI and BG hedge mustard wild populations at two production cycles. Data are expressed as means of three plants ($n = 3$).

| Cycle | Fertilization Level | Wild Population | Fresh Matter (g/plant) | Kg/m$^2$ | Dry Matter (%) |
|-------|---------------------|-----------------|------------------------|----------|----------------|
| I | 100% | MI | 35.5 | 0.25 | 7.6 |
| | | BG | 67.1 | 0.47 | 8.1 |
| | 70% | MI | 31.8 | 0.22 | 7.2 |
| | | BG | 40.7 | 0.28 | 7.8 |
| | 50% | MI | 30 | 0.21 | 7.7 |
| | | BG | 55.1 | 0.38 | 9.1 |
| II | 100% | MI | 33.7 | 0.23 | 13.8 |
| | | BG | 32.8 | 0.23 | 13.4 |
| | 70% | MI | 41.3 | 0.29 | 11.4 |
| | | BG | 28 | 0.20 | 16.2 |
| | 50% | MI | 32.1 | 0.22 | 12.6 |
| | | BG | 25.9 | 0.18 | 14.9 |

Data were subjected to multifactor ANOVA and differences among wild populations and dates within a cycle were determined using LSD's test ($p = 0.05$). The results of the statistical analysis have been reported in the supplementary table (Table S7).

The multifactor ANOVA performed for chl *a+b*, and carotenoids data showed that the only significant differences were found for the growing cycles. The interaction among all factors was significant for $p < 0.05$.

Among the different fertilization levels (100%, 70%, and 50%), no significant differences were found in Chl *a+b*, nor in total carotenoids content (Figure 1A,B).

For Chl *a+b* content, higher values were observed during the second cultivation cycle the where MI wild population showed a higher content at the fertilization level of the 100% 341.21 mg/100 g FM, and it showed a lower level at the 50% 239.04 mg/100 g FM (Figure 1B). Differently, in the BG wild population the during second cultivation cycle, the highest content was recorded at the fertilization level of 70% 309.56 mg/100 g FM, and the lowest level was recorded at 100% 189.49 mg/100 g FM (Figure 1B). During first cultivation cycle, the highest Chl *a+b* concentration was register for the MI

wild population at the fertilization level of 70% and 229.62 mg/100 g FM, and the lowest one was recorded for the same fertilization level in BG wild population, 124.14 mg/100 g FM (Figure 1A).

As observed for total chlorophylls, total carotenoids were higher during the second cultivation cycle, ranging from 58.82 mg/100 g FM for the MI wild population at the fertilization level of 100% to 35.29 mg/100 g FM in the BG wild population at the same fertilization rate (Figure 2B).

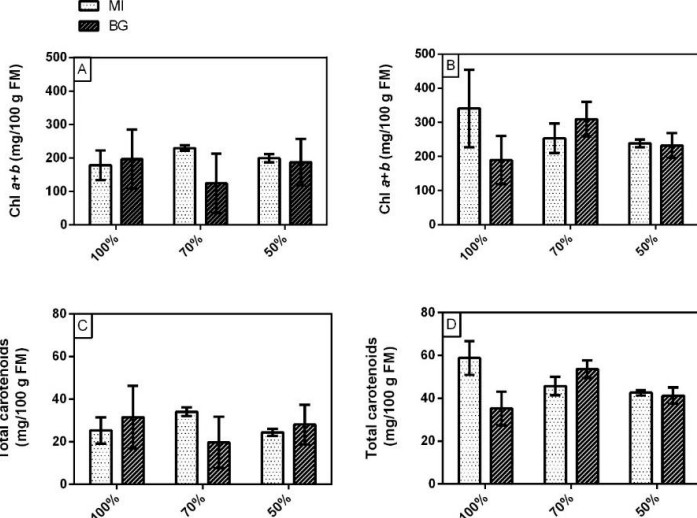

**Figure 1.** Chlorophyll (*a+b*) and total carotenoids content in fresh leaves of two hedge mustard wild populations, MI and BG, cultivated under three different fertilization levels (100%, 70%, and 50%). Graphs (**A**,**C**) refer to the first cultivation cycle, while graphs (**B**,**D**) refer to the second cultivation cycle. Data are means with standard errors (*n* = 3). Data were subjected to multifactor ANOVA, and differences between wild populations were determined using LSD's test (*p* = 0.05).

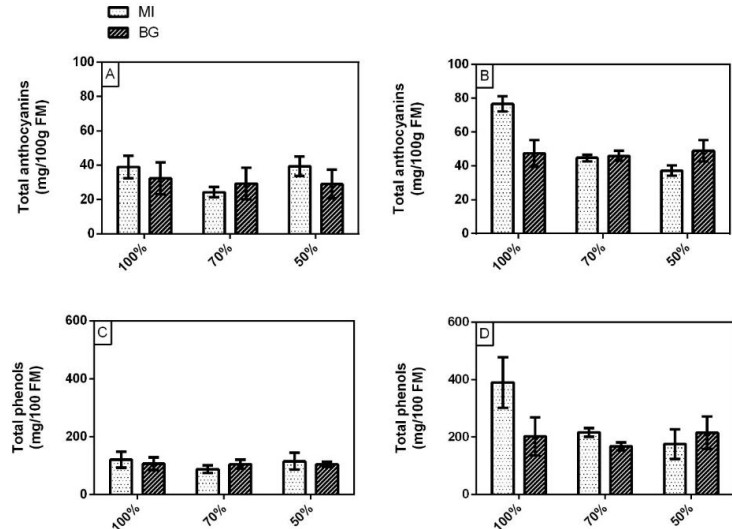

**Figure 2.** Total anthocyanins and total phenols content in fresh leaves of two hedge mustard wild populations, MI and BG, cultivated under three different fertilization levels (100%, 70% and 50%). Graphs (**A**,**C**) refer to the first cultivation cycle, while graphs (**B**,**D**) refer to the second cultivation cycle. Data are means with standard errors (*n* = 3). Data were subjected to multifactor ANOVA, and differences between wild populations were determined using LSD's test (*p* < 0.05). The multifactor ANOVA performed for chl *a+b* and carotenoids data showed that the only significant differences were found for the growing cycles (Tables S8 and S9). The interaction among all factors was significant for *p* < 0.05.

The same trend between first and second cultivation cycles could be also observed for total anthocyanins and phenolic index values in the two *S. officinale* wild populations.

Multifactor ANOVA performed for anthocyanins revealed that significant data were found for fertilization and cycle factors. The interactions between "fertilization rates × wild population" and "fertilization × wild population × cycle" were significant for $p < 0.05$ (Table S8).

During the first cultivation cycle, the total anthocyanins content measured for each fertilization rate was overall lower than that one observed in the second cultivation cycle. The MI wild population at the fertilization level of 50% showed the highest total anthocyanins content, 39.38 mg/100 g FM. Significant differences were found among the three different fertilization levels and wild populations, and, in particular, 50% and 75% fertilization rates were different from 100% (Figure 2A, Table S8). Throughout the second cycle, the same wild population showed the highest content, 76.65 mg/100 g FM, this time at the fertilization rate of 100% (Figure 2B). In this second case, there were significant differences between the MI wild population at the fertilization level of 100% and the other samples.

The multifactor ANOVA for the total phenols showed that all factors and all interactions were statistically different (Table S9).

Phenolic content during the first cultivation cycle ranged between 120.68 mg/100 g FM and 87.83 mg/100 g FM (Figure 3C). During the second cycle, the highest content was observed in the MI wild population sample at the fertilization rate of 100% (390.52 mg/100 g FM), and values ranged between 216.73 and 167.82 mg/100 g FM (Figure 2D).

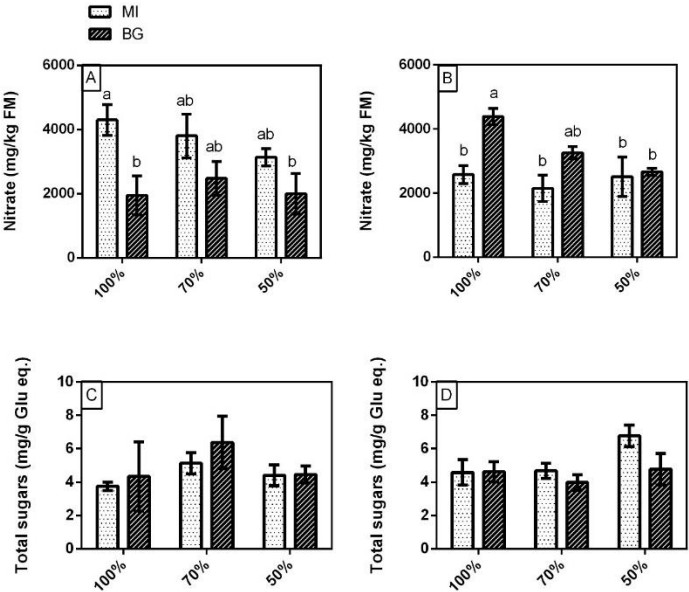

**Figure 3.** Nitrate and total sugars content in fresh leaves of two hedge mustard wild populations, MI and BG, cultivated under three different fertilization levels (100%, 70% and 50%). Graphs (**A**,**C**) refer to the first cultivation cycle, while graphs (**B**,**D**) refer to second cultivation cycle. Data are means with standard errors (*n* = 3). Data were subjected to Multifactor ANOVA, and two-way ANOVA and differences among wild populations were determined using Tukey's test (*p* < 0.05). Different letters indicate statistical differences, while no letters indicate no significant differences. Multifactor ANOVA results have been reported in the supplementary table (Table S10 for Nitrate).

### 3.3. Nitrate and Total Sugars

Nitrate content subjected to multifactor ANOVA showed a significant difference only among the fertilization rates (Table S10). A significant interaction was only found between "wild population × cycle." In fact, during the first cultivation cycle, the MI wild population showed the highest values, while in the second cycle, the BG wild population showed the highest nitrate content. In both cases, it is possible to observe a dose-response to nitrate concentration with the highest value at the fertilization

rate of 100%, and lower one under 50%. Differences were observed between the 50% and 100% fertilization rates.

During the first cultivation cycle, the nitrate content ranged between 4295.58 (MI wild population 100%, Figure 3A) and 1947.21 mg/kg FM (BG wild population 100%, Figure 3A). During the second cultivation cycle, the nitrate content ranged between the highest value, observed in BG 100%—4388.65 mg/kg FM—and the MI 50% wild population, 2148.93 mg/kg FM (Figure 3B).

Statistical differences were observed among the MI wild population at fertilization level of 100% and the BG wild population at 100% and 50% rates for the first cultivation cycle; these differences were also observed among the BG wild population at the fertilization level of 100% and the other samples (apart from sample under the 70% fertilization level) of the same wild population.

The total sugars content did not show significant differences among the three different fertilization levels, and it ranged between 3.74 mg/kg Glu eq. in the first cultivation cycle (MI wild population 100%, Figure 3C) and 6.77 mg/kg Glu eq. in the second cycle (MI wild population 50%, Figure 3D). Multifactor ANOVA showed no significant differences among factors.

### 3.4. Glucosinolates

Glucosinolates (GLS) were investigated as distinguishing secondary metabolites in the Brassicaceae family. The multifactor ANOVA performed on GLS data showed significant differences in the growing cycles. Significant interactions were found between "Fertilization × wild population" and "fertilization × cycle".

During the first cultivation cycle, GLS content ranged from 12.16 μmol/g FM (MI 100%) to 6.39 μmol/g FM (MI 70%). The two *S. officinale* wild populations showed a different behavior in GLS concentration for the three fertilization levels: The MI wild population showed a higher concentration at the fertilization level of 100% and a lower concentration the at 70% and 50% rates, with no statistical differences among the fertilization treatments, while the BG wild population had the opposite response—that is, a lesser GLS concentration under the 100% fertilization level and a higher one under the 50% fertilization level (Figure 4A, no statistical difference).

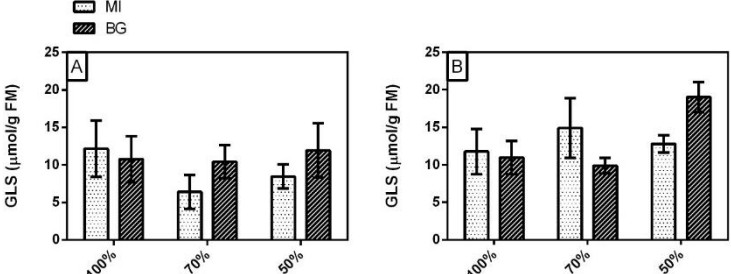

**Figure 4.** Glucosinolates (GLS) content expressed as μmol/g (fresh matter) FM of two hedge mustard wild populations, MI and BG, cultivated under three different fertilization levels (100%, 70% and 50%). Graphs (**A**) refers to the first cultivation cycle, while graph (**B**) refers to the second cultivation cycle. Data are means with standard errors (*n* = 3). Data were subjected to multifactor ANOVA, and differences among fertilization were determined using LSD's test (*p* < 0.05).

In Figure 4B, the GLS concentrations for the second cultivation cycle, ranging from 19 to 9.88 μmol/g FM, are reported. The highest concentration was measured at the 50% fertilization rate in the BG wild population. This treatment was statistically different among the others, even for the same wild population. Moreover, the BG wild population in the three different levels showed the same behavior both during first and second cultivation cycle.

## 4. Discussion

The potential use of *S. officinale*, commonly known for its curative effect on the vocal tract, as a leafy vegetable has been preliminary tested to understand its agronomic and phytochemical potential, as well as its commercialization as minimally processed vegetables.

Nowadays, there are many commercial herbal products containing different parts of hedge mustard plants in combination with other botanicals. As these products are largely used by professional users of the voice, there is a huge interest on understanding the molecular mechanism for its action or its active principles. Several literature reports have confirmed that the TRPA1 ion channel is a polymodal sensor involved in mediating inflammatory pain signals. Plants in the Brassicaceae family are rich in TRPA1 active compounds. Borgonovo et al. found that in vitro essay of isothiocyanate (ITC), isopropylisothiocyante (IPITC) and 2-buthylisothiocyanate (2-BITC) are selective and, in some cases, potent antagonists of TRPA1 channels [34].

Regarding the agronomic aspects, the yield was not significantly affected by the fertilization treatments, at least when this species was grown for baby leaf production (13th BBCH growth stage). These results suggest that the lowest fertilization rate used could be considered as a starting point for further investigations.

The means in the production (FM) of *S. officinale* as leafy vegetables were lower when compared with similar species, such as *E. sativa* (0.7–1.2 kg/m$^2$) and *Diplotaxis tenuifolia* (0.5–1 kg/m$^2$) [35]. This reduction in production can be associated with the shoot architecture, and further studies on plant density are required to define the optimal density for the highest yield.

Leaf pigments were evaluated, as they are related to the visual appearance and the physiological function of leaves. Chlorophylls and carotenoids are pigments involved in light harvesting and energy transmission into the photosynthetic system. In the case of a light energy surplus, for example, carotenoids dissipate the excess energy to prevent photosynthetic damages. Anthocyanins can also act as carotenoids to protect leaves from excess light or from UV radiation and to serve as scavengers of reactive oxygen intermediates or as antifungal compounds. For that reason, the evaluation of leaf pigments content is an important parameter to consider as it gives not only a first quality parameter but also some important information about the physiological state of leaves [36]. However, all these molecules contribute to the antioxidant potential that can be exploited in the human nutrition.

When consumers buy leafy vegetables, the first quality evaluation is based on the color and visual appearance of the product. In the present study, the total chlorophylls (*a+b*) concentration measured in *S. officinale* leaves was in general higher than the values observed in other *Brassica* leafy vegetables, while total carotenoids were similar or slightly higher [37,38].

The second cultivation cycle showed higher values for both chlorophylls and carotenoids content, indicating that there might be a seasonal influence (see Figure S1) on the increase of these pigments.

The results emphasize a different behavior of the two wild populations with respect to the environmental conditions (climate + fertilization level).

The total phenolics content measured in hedge mustard fresh was higher than the one observed in mustard (62.0 ± 1.1 mg/100 g), another species belonging to the Brassicaceae family [39].

Nitrate levels in fresh leaves can vary within species, the cultivar of the same species, and even in genotypes with different ploidy [40]. Rocket is a vegetable species that can accumulate the highest nitrate content (>5000 mg/kg FM), and *Brassica* vegetables like bok choy, Chinese cabbage, mustard greens, and Swiss chard are considered high nitrate accumulating species, where $NO_{3-}$ content ranges between 100 and 2500 mg/kg FW [41].

Normally, in *Eruca sativa* nitrate content ranged between 1528 and 7340, while in *Eruca vesicaria* it ranged between 963 and 4305 mg/kg FM [42]. The environmental conditions can deeply affect the nitrate assimilation pathway. The main environmental parameters that influence the nitrate reduction and incorporation of amino acids are cultivation systems (soil or hydroponics) management, light intensity, photoperiod, temperature, nitrogen availability, and nitrogen forms (nitrate or ammonium). The light intensity and photoperiod are responsible for carbon skeletons biosynthesis (sugars) that

can be used for the incorporation of reduced nitrate. Therefore, higher light intensity and longer photoperiods increase the potential nitrate assimilation ability of crops. The nitrate reduction pathway is regulated by two key enzymes—nitrate and nitrite reductase. Both these enzymes can be regulated by environmental temperature. Hence, for each crop, there is an optimal range of temperatures that accelerates nitrate assimilation. The nitrogen supply as nitrate or ammonium can have effect on the nitrate accumulation in leaves. Nitrogen supplied as ammonium in the hydroponic systems can reduce nitrate accumulation, but the concentration cannot overcome 50% of the total nitrogen supplied, as phytotoxicity can occur [43]. Finally, the most important strategy in soil cultivation is the adequate supply of nitrogen amount. The correct amount avoids the excessive uptake and accumulation in leaves.

Different agricultural systems can have different impacts on the nitrate accumulation in leaves of *E. sativa*, with values that ranging between 1575 and 4139 mg/kg FM under organic cultivation, 2720–6036 mg/kg FM in soil cultivation, and from 4716 to 7083 mg/kg FM in hydroponics [43].

The nitrate content measured in the present study (ranging from 4388.64 mg/kg FM in the 100% BG wild population during the second cultivation cycle to 1947.21 mg/kg FM in the same wild population and fertilization level during the first cultivation cycle) shows the attitude of this species to high-accumulate $NO_{3-}$, even if the level of this nutritive element does not exceed the limits imposed by the EU [22]. The nitrate accumulation on leaves was affected by the dose fertilizer applications. These results suggest that the nitrogen supply should be taken in consideration during cultivation. Further studies should also consider the form of nitrogen supply and identify the optimal ratio between nitrate and ammonium.

Furthermore, the results prove the possibility to reduce nitrate content working on the fertilization level. Fertilization application at levels of 70% and 50% have shown a reduction in nitrate content, especially for the BG wild population in the first cultivation cycle and in the MI wild population for the second one.

The fertilization level at 70% during the first cultivation cycle and at 50% in the last cycle, showed the highest total sugars content, and the values measured were slightly higher than those observed in a previous study on the same species [44].

Glucosinolates are antioxidant compounds that are biosynthesized starting from the methionine with several intermediate steps. The nutrients availability, especially nitrogen and sulfur, have a crucial function in GLS biosynthesis. Broccoli crops grown under different levels of sulfur and nitrogen demonstrated that the reduction of nitrogen availability strongly increased glucosinolates biosynthesis, much more than the sulfur levels [45]. However, the nitrogen availability cannot be reduced under certain levels, otherwise it can affect the chlorophylls concentration of the leaves and the visual appearance of the produce.

The average of GLS as DM (dry matter), considering the 81% of moisture, ranged between 110.11 μmol/g DM in the 50% BG wild population during the second cultivation cycle to 33.67 μmol/g DM, recorded in the 70% MI wild population during the first cultivation cycle. These results were higher than the total amount of GLS found in leaves of other medicinal plants [46].

## 5. Conclusions

*Sisymbrium officinale* is a medicinal plant that belongs to the Brassicaceae family, common in the Euroasiatic area as a wild species. The results of our study suggest that *S. officinale* can be cultivated as a leafy vegetable at the 13th BBCH growth stage. These results also suggest that the use of lower fertilization can enhance some important quality compounds, such as pigments and total sugars and, at the same time, can reduce nitrate content. Results suggest that the best fertilization treatment under the experimental conditions was the 50% fertilization level. Between the two wild populations, significant differences in qualitative and quantitative parameters were observed, and, for that reason, further investigations, such as genotypic characterization, could be desirable.

**Supplementary Materials:** The following are available online at http://www.mdpi.com/2073-4395/9/7/401/s1, Figure S1. Temperature minimum and maximum and solar radiation (SR) during the cultivation period, Table S1. Cultivation period including sowing, transplant, sampling and harvesting time, and the total number of plants per each wild population, Table S2. Fertilization levels (100%, 70%, and 50%) using a NPK granular fertilizer (14:7:17), Table S3. Analysis of Variance for Fv/Fm - Type III Sums of Squares, Table S4. Analysis of Variance for PI—Type III Sums of Squares, Table S5. Analysis of Variance for DIo/RC—Type III Sums of Squares, Table S6. Analysis of Variance for RC/CSm—Type III Sums of Squares, Table S7. Analysis of Variance for DM (Dried Matter)—Type III Sums of Squares, Table S8. Analysis of Variance for Anthocyanins—Type III Sums of Squares, Table S9. Analysis of Variance for Total Phenols—Type III Sums of Squares, Table S10. Analysis of Variance for Nitrate—Type III Sums of Squares.

**Author Contributions:** Conceptualization, A.B. and A.F.; methodology, M.G. and G.B.; formal analysis, M.G. and G.B.; investigation, M.G.; resources, X.X.; data curation, M.G.; writing—original draft preparation, M.G. and A.F.; writing—review and editing, G.B., A.B., and A.F.; visualization, M.G.; supervision, A.F.; project administration, A.B.; funding acquisition, A.B.

**Funding:** This research was funded by Fondazione Cariplo, ERISIMO A MILANO project, no. 2017-1653.

**Conflicts of Interest:** The authors declare no conflict of interest.

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
