# Peer review of "Effect of Fertilization on Yield and Quality of Sisymbrium officinale (L.) Scop. Grown as Leafy Vegetable Crop"

_agronomy, doi:10.3390/agronomy9070401_

Round 1
Reviewer 1 Report
The publication presented for the review contains very interesting information about the examined plant. It is known that wild plants - often herbs can be used in phytopharmacy.
I marked my comments in the comments (Yellow windows). Especially, please use the BBCH plant scale to determine the date of harvesting and testing leaf samples.

Author Response
All authors would like to thank you the reviewer for the important corrections that they are certain will improve the manuscript in its entirely and clarity.
The publication presented for the review contains very interesting information about the examined plant. It is known that wild plants - often herbs can be used in phytopharmacy.
I marked my comments in the comments (Yellow windows). Especially, please use the BBCH plant scale to determine the date of harvesting and testing leaf samples.
A.A: The BBCH plant scale was used to determine the date of harvesting and sampling as requested.
We considered also the other revisions and corrected them.
Reviewer 2 Report
The manuscript is unnecessary too long because of several superfluous details provided by the authors. For example, the authors could well do away with the paragraph in lines 57-64 and still accurately present the rationale of the study. Similarly, paragraph in lines 80-92 is really not needed given that the authors did not examine the molecular mechanism of the compounds found in these species. Lines 151-155 seem like a repeat of the materials and methods section. In essence, the introduction has to be shortened and improved. Also there are few grammatical errors across the entire manuscript.
Please clearly lay out the treatment structure (experimental design, number of replication used).
Why were the levels of P2O5 and K2O not adjusted to the same across the three N rates evaluated? There is a possibility of confounding effect of N, P, and K on the reported results.
Line 194: What are these qualitative characteristic that were evaluated?
Statistical analyses: Why was 'the cultivation cycle' not included as a factor? Maybe, the authors should consider analyzing the pooled data from the cycles instead of separately especially if they don't want to include 'cycle' as a factor in the model. Mention the statistical software used.
The results section requires significant improvements by first eliminating extraneous comments (lines 261-262, 285-286, just to name few). Also, given the small number of samples evaluated (n=4 in table 1), it is advisable to combine data of the cycles for the analysis. Lines 276-277 and 281 are repeated.
Since this was a two-way ANOVA, the significance of the interaction between the two factors has to be discussed first. Depending on whether there is a significant effect of the interaction or not, the main effect or simple effect of these factors should be presented.
The results and discussion sections are marred with several grammatical errors that need to be addressed.
In conclusion, the manuscript can be accepted after major revision.
Author Response
All authors would like to thank the reviewer for the important corrections that we are certain will improve the manuscript in its entirely and clarity.
The manuscript is unnecessary too long because of several superfluous details provided by the authors. For example, the authors could well do away with the paragraph in lines 57-64 and still accurately present the rationale of the study.
A.A: as suggested the introduction has been reduced even if some parts have been preserved since they are important for the understanding of the aim of the study.
Similarly, paragraph in lines 80-92 is really not needed given that the authors did not examine the molecular mechanism of the compounds found in these species.
A.A: Lines 80-92 were removed from the introduction and partly reported in the conclusion.
Lines 151-155 seem like a repeat of the materials and methods section. In essence, the introduction has to be shortened and improved. Also there are few grammatical errors across the entire manuscript.
A.A: Lines 151-155 were removed, and grammatical errors were corrected.
Please clearly lay out the treatment structure (experimental design, number of replication used).
A.A: The experimental design was organized as follow:
- two wild population (MI and BG);
- three different fertilization levels (100%, 70%, and 50%);
- six plants for each fertilization levels for a total number of 18 plants for MI wild population, and 18 for BG wild population;
- two cultivation cycles.
Below the number of replications performed:
- four replicates (n=4) for each fertilization level of each wild population for the evaluation of chlorophyll a fluorescence;
- three replicates (n=3) for each fertilization level of each wild population for the other qualitative parameters that we decided to consider.
All these information are reported in the Statistical section.
Why were the levels of P2O5 and K2O not adjusted to the same across the three N rates evaluated? There is a possibility of confounding effect of N, P, and K on the reported results.
A.A: The fertilization was defined considering rape as reference species for the nutrient requirements that is a crop with long growing cycle, therefore the amount of fertilizers must be lowered since the Erisimo was grown as baby leaf vegetable. However, the ratio N:P:K among nutrients should be similar to the rape therefore the reduction was performed for all nutrient keeping the same ratio.
Line 194: What are these qualitative characteristic that were evaluated?
A.A: The chlorophyll a fluorescence evaluates the functionality of the leaf and therefore the state of health (or stress) of the entire plant.
The light energy absorbed by the leaf excites the electrons in the chlorophyll molecules. The energy in photosystem II can converted into chemical energy to drive photosynthesis (photochemistry). If the photochemical reaction is inefficient, the excess energy can be emitted:
1) in the form of heat (non-photochemical quenching);
2) re-emitted as chlorophyll fluorescence (1-2%).
By measuring the amount of chlorophyll a fluorescence, photochemistry efficiency and non-photochemical quenching can be derived.
Handy PEA instrument was used to measure some important parameters relate to PSII efficiency, other data were calculated starting from those detecting by the instrument.
Among all these data, only four parameters, which we think can be more significant, were considered:
-Fv/Fm, which shows the maximum quanticum efficiency of PSII;
-PI, an index of the PSII performance. For the majority of the plants, PI value in non-stressful condition is 0.83, lower values suggest conditions of stress for the plant and photosystemic apparatus;
-DIo/RC, which shows the dissipation of energy for reaction centers;
-RC/CSm, which suggests the density of the reaction centers.
Statistical analyses: Why was 'the cultivation cycle' not included as a factor? Maybe, the authors should consider analyzing the pooled data from the cycles instead of separately especially if they don't want to include 'cycle' as a factor in the model. Mention the statistical software used.
A.A: Considering the reviewer’s comments, the Mutifactor ANOVA has been applied including the cycle as factor. All the statistical analyses have been reported as supplementary data and the results section has been revised according to the multifactor ANOVA results.
The results section requires significant improvements by first eliminating extraneous comments (lines 261-262, 285-286, just to name few). Also, given the small number of samples evaluated (n=4 in table 1), it is advisable to combine data of the cycles for the analysis. Lines 276-277 and 281 are repeated.
A.A: Lines 276-277 and 281 were corrected and moved to the right position in the text.
Since this was a two-way ANOVA, the significance of the interaction between the two factors has to be discussed first. Depending on whether there is a significant effect of the interaction or not, the main effect or simple effect of these factors should be presented.
A.A: As reported above the Multifactor ANOVA has been reported.
The results and discussion sections are marred with several grammatical errors that need to be addressed.
A.A.: the entire manuscript has been revised and grammatical mistakes corrected.
Reviewer 3 Report
The study is very practical and meaningful. However I had some major and minor comments.
Major
· Sampling procedure and statistical analysis should be clarified. If I understand right sampling for chlorophylls, phenolic was done at the time of harvest. About nitrates sampling was done twice, two times during the first cycle and once during the second without any more information’s. Why twice in the 1st cycle and when? On the other hand for sugars sampling was done 3 times, no further information’s are provided. For GLS no sampling information’s are given. Moreover, in all graphs means are the average of n=3. Please clarify all the above
· Why the difference of two month for the establishment of the experiment between the 2 cycles? Maybe the different environmental conditions could alter the 2nd cycle as replication of the experiment?
· Authors should discuss in more depth their results of nitrate concentration. It is well known that nitrate concentration is decreased during period of excess sunlight than the winter period. For BG population in the 2nd cultivation cycle (later harvest during summer) in all fertilization levels nitrates was increased. Especially in 100% the nitrate concentration was increased more than 100%. On the other hand MI population had the expected pattern of nitrate concentration with decrease during the 2nd cultivation period.
· Delete lines 151-158 they are Materials and Methods
Minor
· Axis scale among the two cultivation cycles for the same parameter should be the same. That will help the reader to understand better the differences among the fertilization level and cultivation cycle eg same pattern
· Usually in wild populations we have some problems with seed germination. Please provide us with some information’s about germination percentage.
· What about the volume of the pots?
Author Response
All authors would like to thank the reviewer for the important corrections that we are certain will improve the manuscript in its entirely and clarity.
Major
· Sampling procedure and statistical analysis should be clarified. If I understand right sampling for chlorophylls, phenolic was done at the time of harvest. About nitrates sampling was done twice, two times during the first cycle and once during the second without any more information’s. Why twice in the 1st cycle and when? On the other hand for sugars sampling was done 3 times, no further information’s are provided. For GLS no sampling information’s are given. Moreover, in all graphs means are the average of n=3. Please clarify all the above
A.A: We apologize for the mistakes in the Materials and Methods section. The mistakes were corrected, and sampling procedures were better explained. Samples (n=3) were collected only at 13th BBCH growth stage (commonly consider as baby leaf stage) only once for each cultivation cycle.
GLS sampling information were added. Only for chlorophyll a fluorescence, parameters were measured before the harvest and in number of four (n=4).
· Why the difference of two month for the establishment of the experiment between the 2 cycles? Maybe the different environmental conditions could alter the 2nd cycle as replication of the experiment?
A.A: As requested also by the reviewer 1 the Multifactor ANOVA has been applied considering as factor also the cycle as suggested by the reviewer. All the statistical analyses have been reported as supplementary data and results have been revised according to the multifactor ANOVA results.
Authors should discuss in more depth their results of nitrate concentration. It is well known that nitrate concentration is decreased during period of excess sunlight than the winter period. For BG population in the 2nd cultivation cycle (later harvest during summer) in all fertilization levels nitrates was increased. Especially in 100% the nitrate concentration was increased more than 100%. On the other hand MI population had the expected pattern of nitrate concentration with decrease during the 2nd cultivation period.
A.A: The nitrate results have been revised and discussed improved as suggested by the reviewer.
· Delete lines 151-158 they are Materials and Methods
A.A: Lines 151-158 were deleted from the introduction section and added to “Materials and methods” section.
Minor
Axis scale among the two cultivation cycles for the same parameter should be the same. That will help the reader to understand better the differences among the fertilization level and cultivation cycle eg same pattern
A.A: The axis scale has been modified.
Usually in wild populations we have some problems with seed germination. Please provide us with some information’s about germination percentage.
A.A: Germination trial was performed using 100 seeds, randomly chosen, for each of the twelve experimental plots at Fondazione Minoprio, one of the project partners, during last summer.
The test gave us important information about the germination rate, which ranged between 94.67 % and 98%.
What about the volume of the pots?
A.A: The volume of the pots used for the experiment was of 2 L. This information was added in the manuscript.
Round 2
Reviewer 2 Report
It would be better to submit a revised version without all the track changes which make it a little difficult for reading. I'm wondering if all the 15 tables in supplement are needed. There are still some grammatical errors. Sentence in line 479 on page 6 seems missing something.
Please mention the statistical software used to analyze the data.
Otherwise, I'm glad that the authors have tried to revise the manuscript based on the suggestions provided.
Author Response
It would be better to submit a revised version without all the track changes which make it a little difficult for reading. I'm wondering if all the 15 tables in supplement are needed. There are still some grammatical errors. Sentence in line 479 on page 6 seems missing something. Please mention the statistical software used to analyze the data.
We thank the reviewer for minor revisions and suggestions. We cordially send the answers to the corrections made.